# Broadband and filter radiometers at Ross Island, Antarctica: Detection of cloud ice phase versus liquid water influences on shortwave and longwave radiation

Kristopher Scarci[1,2], Ryan C. Scott[3], Madison L. Ghiz[4], Andrew M. Vogelmann[5], and Dan Lubin[1]

[1]Scripps Institution of Oceanography, University of California San Diego, La Jolla CA, 92093-0206, USA
[2]18508 114th Dr. NE, Arlington WA, 98223, USA
[3]NASA Langley Research Center, Hampton VA 23666, USA, now at BlackSky Technology, Herndon VA, 20171, USA
[4]DNV Inc., San Diego CA 92123, USA
[5]Environmental and Climate Sciences Department, Brookhaven National Laboratory, Upton NY 11973-5000 USA

Correspondence to: Dan Lubin (dlubin@ucsd.edu)

**Abstract.** Surface radiometer data from Ross Island, Antarctica, collected during the austral summer 2015-16 by the US Department of Energy Atmospheric Radiation Measurement (ARM) program West Antarctic Radiation Experiment (AWARE), are used to evaluate how shortwave and longwave irradiance respond to changing cloud properties as governed by contrasting meteorological regimes. Shortwave atmospheric transmittance is derived from pyranometer measurements, and cloud conservative-scattering optical depth is derived from filter radiometer measurements at 870 nm. With onshore flow associated with marine air masses, clouds contain mostly liquid water. With southerly flow over the Transantarctic Mountains, orographic forcing induces substantial cloud ice water content. These ice and mixed-phase clouds attenuate more surface shortwave irradiance than the maritime-influenced clouds, and also emit less longwave irradiance due to colder cloud base temperature. These detected irradiance changes are in a range that can mean onset or inhibition of surface melt over ice shelves. This study demonstrates how basic and relatively low-cost broadband and filter radiometers can be used to detect subtle climatological influences of contrasting cloud microphysical properties at very remote locations.

## 1 Introduction

Even with the numerous recent advances in atmospheric measurement technology, there are many remote regions on Earth that can be sampled with only very basic and fully automated instruments. Such locations naturally include much of the high latitudes, but also include isolated islands or sea-based platforms, remote forest, desert or alpine locations, or locations where atmospheric observations are made as ancillary data to some other environmental monitoring priority. The purpose of this paper is to demonstrate what can be learned about a region's climate from basic atmospheric irradiance measurements made by pyranometers, pyrgeometers and filter radiometers, with the aid of meteorological reanalysis and satellite remote sensing.

The Antarctic is one critical region for terrestrial climate change whose extreme remoteness and numerous logistical challenges have led researchers to rely heavily on networks of basic instruments such as automatic weather stations (AWS, Lazzara et al. 2012). In one example, shortly after the collapse of the Larsen-B Ice Shelf (Glasser and Scambos, 2008) an AWS provided the only in situ data conclusively demonstrating the effect of a circulation anomaly that transported warm air

directly to the ice shelf surface, spurring melt-ponding, hydrofracturing, and the rapid ice shelf collapse (van den Broeke, 2005). In another example, pyranometers deployed aboard the Australian research vessel RSV *Aurora Australis* throughout several years of Southern Ocean transects were analyzed in conjunction with shipboard sky and sea ice observations, and detailed radiative transfer modeling, to derive valuable climatological cloud optical and radiative properties over sea ice and open ocean (Fitzpatrick et al., 2004; Fitzpatrick and Warren, 2005; 2007).


Since these earlier studies the role of Antarctica in cryospheric mass loss and sea level rise has seen increasing attention. Between 1979-2017 the total mass loss from the Antarctic continent has increased by a factor of six, and most recently this mass loss has been dominated by the Amundsen/Bellingshausen Sea sectors of West Antarctica and Wilkes Land in East Antarctica (Rignot et al., 2019). The main concern about the stability of the West Antarctic Ice Sheet (WAIS) has centered

on its response to a warming Southern Ocean in the form of ice-sheet-grounding-line retreat inland on downward slopes, a mechanically unstable system that potentially results in large-scale loss of WAIS ice mass if the warming process reaches a tipping point (Alley et al., 2015). At the same time, the adjacent floating ice shelves comprise a critical buffer to ice sheet loss by being anchored to adjacent land formations, thereby providing mechanical buttressing. This ice sheet buttressing occurs substantially throughout all the West Antarctic ice outflows of most critical concern in today's warming climate

(Fürst et al., 2016). Recent ice sheet modeling (Pollard et al., 2015) and observational studies (e.g., Scambos et al., 2009; McGrath et al., 2015) emphasize that surface melt on ice shelves - induced by atmospheric warming - can potentially accelerate mass loss in West Antarctica and on the Antarctic Peninsula. At high latitudes during summer the radiative fluxes are generally the largest individual components of the surface energy balance (SEB), and radiative fluxes over West Antarctica are strongly modulated by cloud variability, including changes in microphysical and optical properties (Scott et

al., 2017). Determining the cloud influences on the SEB therefore has increasing relevance for understanding cryospheric change. Specifically, changes in net surface radiative flux of order 10 W m$^{-2}$ or less can make the difference between a SEB having a positive-negative diurnal cycle, or remaining positive for several days and inducing or sustaining surface melt (Nicolas et al., 2017; Ghiz et al., 2021).

These issues, ultimately related to sea level rise, in 2015 motivated the US Department of Energy (DOE) Atmospheric Radiation Measurement (ARM) program and the US Antarctic Program (USAP) to jointly deploy an advanced and comprehensive suite of atmospheric science equipment to Antarctica (Lubin et al., 2020). This campaign, the ARM West Antarctic Radiation Experiment (AWARE), installed the Second ARM Mobile Facility (AMF2; Mather and Voyles, 2013) at McMurdo Station on Ross Island (77.85°S, 166.66°E), where it operated over a full annual cycle with all instruments

starting on 1 December 2015 and ending on 5 January 2017. AMF2 instrumentation included a set of triple-frequency cloud research radars, cloud and aerosol polarized lidars, meteorological equipment, rawinsondes, aerosol microphysical and chemical sampling equipment, and a variety of upward- and downward-looking radiometers. The AMF2 was installed at the site of the USAP Cosmic Ray Laboratory (CosRay), in an open area approximately 1 km from McMurdo Station and along the main road leading to New Zealand's Scott Base. A second component of AWARE deployed a smaller suite of instruments optimized for SEB measurement to the WAIS Divide Ice Camp (79.47°S, 112.09°W) where it operated from 4 December 2015 until 17 January 2016. At WAIS Divide the AWARE campaign was able to sample the SEB during a major surface melt event, directly related to the cryospheric change issues discussed above, including the role of cloud microphysical properties (Nicolas et al., 2017; Wilson et al., 2018). Here we analyze radiometer data from the main AMF deployment at the CosRay site on Ross Island, encompassing 91 days of measurements from the austral summer 2015-2016. A map of Antarctica showing the AWARE deployments and other locations discussed in this paper is shown in Figure 1.

Data from AWARE's most sophisticated instruments have now been extensively analyzed in climatological studies that include contrasts in cloud microphysical properties between the Antarctic and Arctic (Zhang et al., 2019), and response of cloud microphysics and radiation to changes in the prevailing meteorological regimes (Silber et al., 2019). The challenge for polar research, especially in the Antarctic, is that the most sophisticated instruments usually can't be deployed at very many locations or for long periods of time. For the foreseeable future, it will be necessary to incrementally build atmospheric science capabilities with relatively simple instruments that can operate long-term at a variety of locations.

Here we analyse data from three basic and well-established ARM-Program instruments for radiation and cloud properties that have long-and-established field records at many sites worldwide. We show how these data sets, combined with meteorological and satellite remote sensing analysis, can detect how significant cloud microphysical changes can result in statistically significant variations in surface shortwave radiation. Our first objective is to determine the prevailing meteorological regimes bringing contrasting air masses to Ross Island during summer 2015-2016. We then analyse ARM radiometer data to determine corresponding contrasts in surface shortwave irradiance related to changing cloud properties. NASA Moderate-resolution Imaging Spectroradiometer (MODIS) retrievals of cloud thermodynamic phase and geometric top height over Ross Island help to explain and corroborate the surface irradiance contrasts we detect in the ARM radiometer data.

### 1.1 Data

We use two standard ARM Program radiometric measurements. The Sky Radiometers on Stand for Downwelling Radiation (SKYRAD; e.g., Long & Ackerman, 2000; Dong et al., 2010) measures downwelling surface shortwave irradiance using Eppley Precision Spectral Pyranometers having a broadband response over the wavelength range 295-3000 nm. SKYRAD measures downwelling longwave irradiance using Eppley Precision Infrared Radiometers having a broadband response over

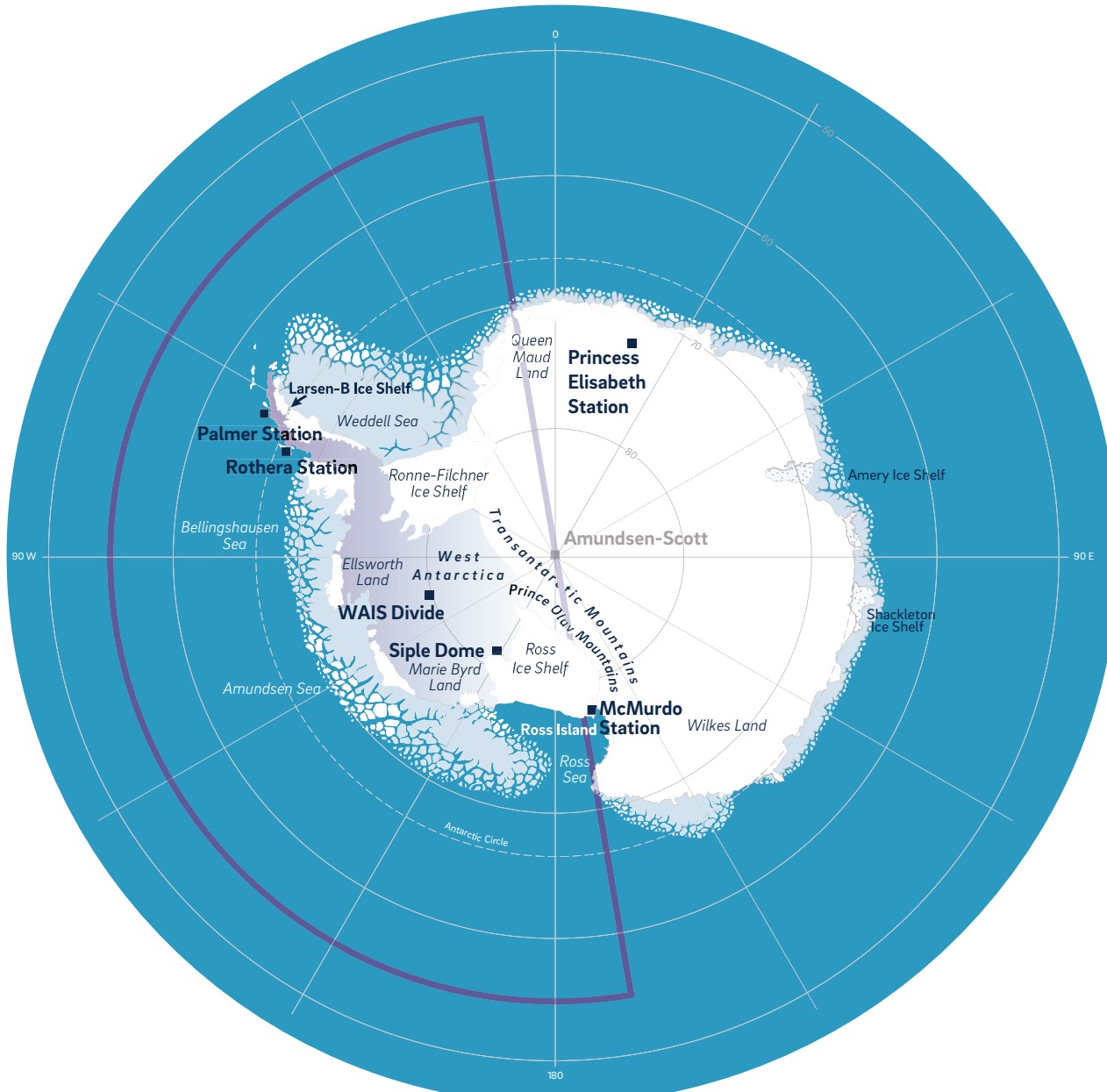

**Figure 1: Map of Antarctica showing the locations of AWARE at McMurdo Station on Ross Island, and WAIS Divide, along with other research stations and geographic regions mentioned throughout the text. The semicircle denoted by the magenta border encloses the geographic region over which ERA-Interim data were analyzed in the k-means clustering algorithm.**

the wavelength range 3.5-50 μm. A complementary ARM suite for upwelling shortwave and longwave radiation reflected and emitted by the surface (GNDRAD) uses the same instruments in a downward-looking configuration. The Multi-Filter

Rotation Shadowband Radiometer (MFRSR, Harrison et al., 1994; Michalsky and Long, 2016) measures narrowband surface irradiance suitable for retrieving cloud scattering optical depth. The standard MFRSR, manufactured by Yankee Environmental Systems (Inc.), comprises six narrowband filter-detector combinations with nominal wavelengths 415, 500, 615, 673, 870 and 940 nm. This wavelength combination allows for retrievals of aerosol and cloud optical depth, total column ozone abundance and total column water vapor. The use of an automated rotating shadowband allows measurement

of the global hemispheric irradiance and the diffuse irradiance, from which the direct solar irradiance is obtained by subtraction. A complementary ARM instrument for upwelling irradiance at these wavelengths, the Multifilter Radiometer (MFR), without a shadowband, allows us to measure the spectral surface albedo for use with radiative transfer calculations. Here we use the 870-nm irradiance from MFRSR and MFR for cloud optical depth retrieval.

We also use the Total Sky Imager (TSI; e.g., Kassianov et al., 2005; 2011) to identify the periods of overcast sky conditions that we interpret in this study. The TSI, also manufactured by Yankee Environmental Systems (Inc.), uses a downward-looking CCD colour camera viewing a concave mirror to image the entire sky with effective field of view 160º. The ARM TSI algorithm, applied when the sun elevation is greater than 10° above the horizon, splits the cloud fractional coverage measurements into two variables, *percent_opaque* and *percent_thin*. We sum these values to determine the total cloud cover,

and classify the scene as overcast if this sum is greater than or equal to 95%.

In the spirit of this study's objective to obtain maximum climatological information from simple yet rugged instruments deployed at very remote locations, we note three aspects of the instruments chosen here. First, pyranometers and pyrgeometers have been used worldwide for many decades. Second, the MFRSR is representative of several emerging

shortwave measurement capabilities adopted by the ARM Facility for long-term observations (Riihimaki et al., 2021). Finally, the TSI itself is generally not an instrument that can be left unattended at remote locations. It normally requires standard AC power and the relatively large concave mirror requires regular cleaning. However, there are now compact, low-power and fully automated digital all-sky imagers such as those manufactured by ALCOR System[TM]. One of these cameras was successfully deployed at Siple Dome in West Antarctica during the austral summer 2019-20 (Lubin et al., 2023).

Therefore the combination of all-sky imagery with broadband and narrowband radiometry is viable for remote deployments and, with careful analysis aided by meteorological reanalysis and satellite remote sensing data, can reveal much about a location's atmospheric properties.

## 2 Meteorological Regimes Influencing Ross Island

Subtle changes in the Antarctic SEB can trigger or sustain surface melt. As clouds are the largest modulator of Antarctic

surface radiation, it is useful to identify prevailing meteorological regimes in a given Antarctic region and determine their associated cloud properties. Scott et al. (2019) performed a *k*-means clustering analysis for the West Antarctic Amundsen Sea Embayment region, home to many of the most vulnerable ice sheets and ice shelves. That study identified nine

characteristic meteorological regimes that can occur in all seasons and either favour or inhibit surface melt by virtue of relatively small differences in atmospheric thermodynamic and cloud properties. Here we use the *k*-means clustering

technique of Mülmenstädt et al. (2011) to objectively identify the prevailing meteorological regimes impacting Ross Island. The observations used as input data for this technique are the ERA-Interim reanalysis (Dee et al., 2011) 700 hPa geopotential height (Z700) anomalies, and 2-m near-surface air temperature (T2m) anomalies, covering a sector of Antarctica and the Southern Ocean ranging from 55°S to the South Pole and from 160°E to 340°E, thus including the entire Ross Ice Shelf (RIS) and WAIS.


The anomalies were obtained by computing daily mean fields from the six-hourly (0000, 0600, 1200, 1800 UTC) fields of Z700 and T2m over a full ten-year period leading up to and including AWARE, from 2007-2016. Then by averaging them for each calendar day, the raw seasonal cycle at each grid point was computed. This time series was then smoothed using fast Fourier transform (FFT) to remove the high frequencies, followed by inverse FFT to get a smoothed seasonal cycle. Daily

anomalies were computed by subtracting the smoothed seasonal cycle from the raw daily means at each grid point. A *k*-means clustering algorithm was then applied to identify characteristic Z700 centroids and to composite the T2m anomalies. These anomalies were computed for each of four seasons (three months periods), and in each season four stable clusters could be identified, each related to a specific orientation of low- versus high-pressure centers in the Ross, Amundsen and Bellingshausen Seas.


During summer (DJF), two clusters are associated with warm meteorological regimes, a third cluster is associated with cold conditions, and a fourth describes a unique large-scale circulation pattern. The corresponding ERA-Interim-based 2 m near-surface temperature anomalies for each cluster (regime) are shown in Figure 2. Regime 1 is characterized by a blocking high over the Amundsen Sea that drives marine air advection and föhn wind-induced warming of Marie Byrd Land and the RIS.

Similar forcing triggered the January 2016 large-scale surface melt event recorded by AWARE data (Nicolas et al., 2017). In this regime, northerly to northeasterly flow brings warm temperatures and low-level clouds to Ross Island of the typical cold maritime stratiform variety. The most likely airmass source region for these clouds is the Ross and Western Amundsen Seas and the eastern Ross Ice Shelf.

Regime 2 is characterized by a low-pressure center over the Bellingshausen Sea that draws continental polar air northward, promoting cooling of West Antarctica (cf. Nicolas & Bromwich, 2011). At the same time, föhn wind warming occurs on the eastern AP. Regime 2 is the "cold" regime impacting Ross Island. Associated with a positive Southern Annular Mode (SAM) index, it favors the coldest average summertime conditions. When present, clouds over Ross Island likely contain significant ice water.


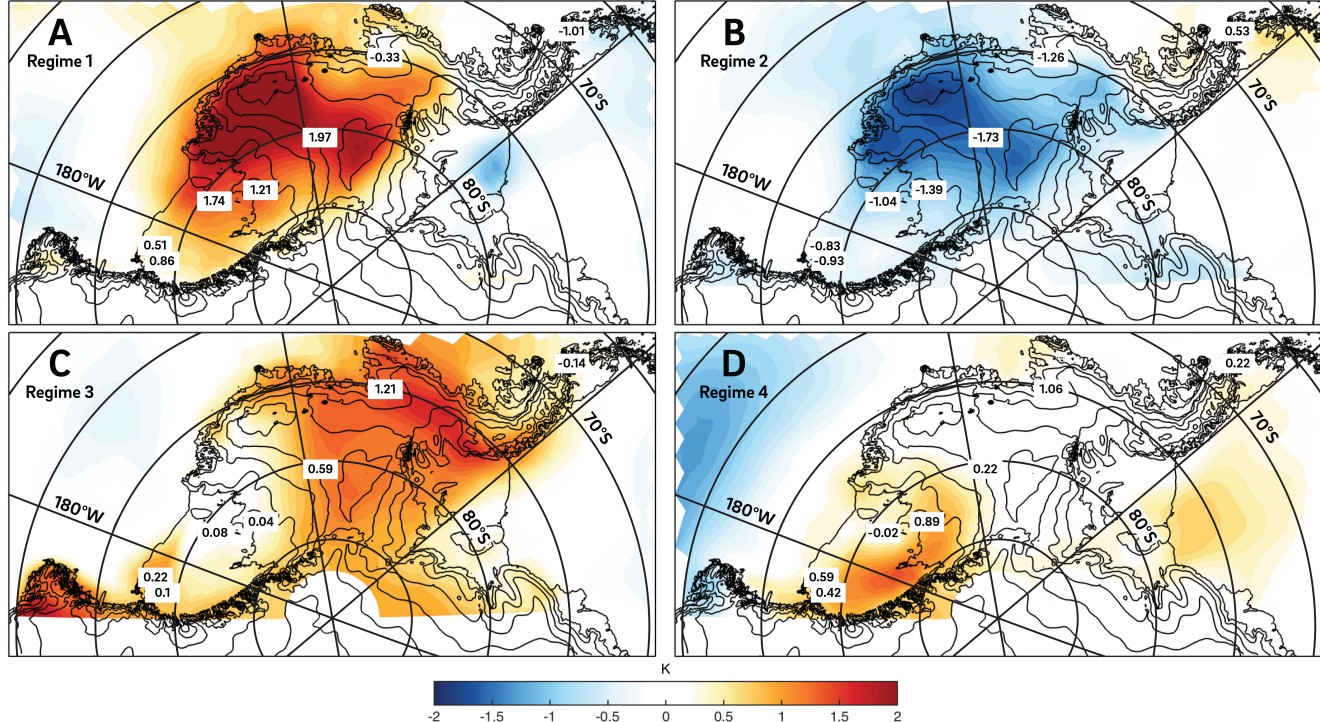

**Figure 2: Anomalies of ERA-Interim-based 2 m near-surface air temperature (color bar) for each of the four meteorological regimes identified by the *k*-means clustering technique for the summer months (DJF). Numbers overlaid on these maps are observed 2 m air temperature anomalies computed from six University of Wisconsin automatic weather stations (Lazzara et al. 2012). Solid lines depict topography in steps of 500 m.**

Regime 3 is driven by a strong negative phase of the SAM and a weak Amundsen Sea low. Two ridging centres over the Ross Sea and the AP favour warming along their western flanks, and moderate föhn warming in Ellsworth Land. Similar but not identical to Regime 1, anticyclonic flow over the Ross Sea advects marine air from the western Ross Sea directly toward Ross Island.

Regime 4 describes a unique circulation pattern in which a deep Ross Sea cyclone injects warm, moist air over Marie Byrd Land, which subsequently descends onto the southern RIS, downwind of the Prince Olav Mountains. Cyclonic intrusions of marine air support a well-developed marine cloud band over West Antarctica. At Ross Island, strong southerly to southeasterly winds prevail throughout the lower troposphere, bringing ice and mixed-phase cloud systems influenced by local and remote orographic forcing, the latter induced by the Transantarctic mountains (Scott et al., 2017).

Figure 2 contains markers that specify 2 m air temperature anomalies computed from Antarctic automatic weather station (AWS) data (Lazzara et al., 2012), and, through their consistency with the ERA-Interim-based temperature anomalies, these independent measurements provide additional confidence in the *k*-means clustering results. ERA-Interim has been superseded by ERA5 (Hersbach et al., 2020) and it is worth considering if differences in the dynamical fields between the two reanalyses potentially give different k-means clustering results. In Figure 3 we compare the 700 hPa geopotential heights between the two for a region over the eastern Ross Sea and RIS. The Pearson correlation between the two is $R = 0.9973$, and there is a slight low bias of order 10 m in ERA-Interim compared with ERA5. This tight consistency between the two reanalysis suggests that the same four clusters would emerge if the work were redone with ERA5. During the summer months sampled by the AWARE campaign on Ross Island when skies were overcast, Regimes 1-4 occurred with frequencies of 25.7, 32.9, 7.1 and 34.3 percent, respectively, and their specific days of occurrence are given in Table 1.

Scott et al. (2019) applied the *k*-means clustering technique to ERA-Interim data over a larger domain to derive nine synoptic patterns influencing surface melt in West Antarctica. Our Regime 1 here is analogous to Pattern 7 of Scott et al. (2019), which favors surface melt. Our cold Regime 2 here is similar to Pattern 2 of Scott et al. (2019). Our Regime 3, which as a small impact on temperature anomalies on the RIS but features warming and melting on the Ronne-Filchner Ice Shelves, is similar to Pattern 5 of Scott et al. (2019). Our unique Regime 4 has similarities with Patter 3 of Scott et al. (2019), particularly regarding warm temperature anomalies over the southern RIS, although in this analysis the driving cyclone appears stronger and situated slightly west of the cyclone derived in Scott et al. (2019). Similarly, Silber et al. (2019) applied a self-organizing map (SOM) technique to classify synoptic regimes influencing Ross Island during AWARE. Our Regimes 1-4 are similar to the Silber et al. (2019) SOM numbers 4, 3, 7 and 6, respectively.

The MODIS cloud products have within their retrieval algorithms a history of increasingly specialized improvements for polar regions (e.g., Frey et al., 2008; Marchant et al., 2016). MODIS cloud property retrievals provide insight into our meteorological regime contrasts. Figure 4a shows the 1-km cloud phase retrievals (Platnick et al., 2003) in a 1° latitude by 1° longitude grid cell over McMurdo Station during summer 2015-2016. Regime 1 shows the largest incidence of liquid water cloud and smallest incidence of mixed-phase cloud. Similarly regime 3 shows a large incidence of liquid water cloud and the smallest incidence of ice cloud. Both results are consistent with onshore flow and marine air mass origin. Regime 2 shows ice or mixed-phase cloud 47% of the time, compared with 27% for liquid water. Regime 4 shows the largest incidence of ice cloud, and ice plus mixed phase cloud comprise 62% of the scene identification compared with 22% for liquid water cloud and 16% for clear sky. MODIS cloud top heights (Figure 4b) show that the two warmer regimes have consistently lower clouds. Regime 4 has the largest range in cloud top height, consistent with orographic forcing by local terrain and the Transantarctic Mountains. Cloud top heights for the cold continental outflow regime are intermediate between the orographic and the two warmer regimes.

**Table 1: Daily occurrences of each *k*-means clustering meteorological regime throughout the austral summer 2015-16.**

| | December 2015 | January 2016 | February 2016 |
|---|---|---|---|
| Regime 1 | 9, 10, 11, 12, 13, 20, 21, 22, 23, 24, 25, 26, 27 | 6, 7, 8, 9, 10, 11, 12, 13, 14, 15 | none |
| Regime 2 | 1, 2, 3, 8, 18, 19, 28, 29, 30 | 20, 21, 22, 23, 24, 25, 26, 28, 29 | 8, 11, 12, 13, 14, 15, 16, 17, 18, 19, 20, 21 |
| Regime 3 | 14, 15, 16, 17, 31 | 1, 2, 3 | none |
| Regime 4 | 4, 5, 6, 7 | 4, 5, 16, 17, 18, 19, 30, 31 | 1, 2, 3, 4, 5, 6, 7, 9, 10, 22, 23, 24, 25, 26, 27, 28, 29 |

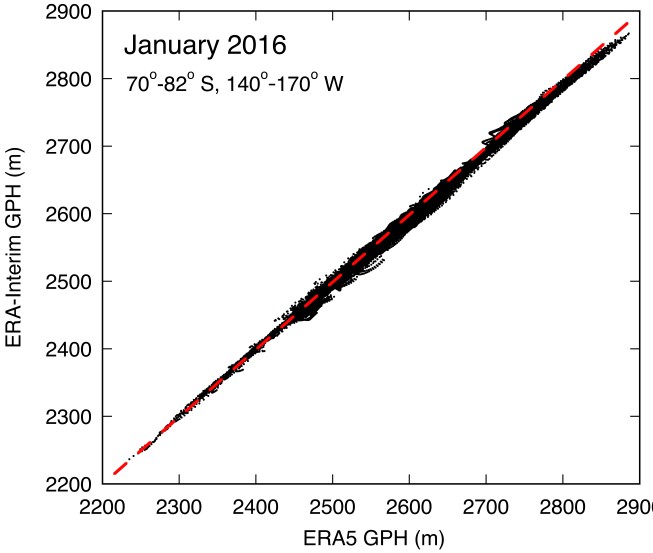

**Figure 3: Geopotential heights at 700 hPa for ERA-Interim versus ERA5 reanalyes for an area over the eastern Ross Sea and Ross Ice Shelf. The dashed red line is the one-to-one correspondence.**

Although MODIS cloud property retrievals have shown improvements over successive data collections, some uncertainties
remain over high latitude regions that may never be fully mitigated. Cossich et al. (2021) compared MODIS cloud amount retrieval over the Antarctic Plateau with retrievals from surface-based spectral infrared remote sensing, and found that MODIS consistently underestimated cloud amount by a factor of two or more year-round, although MODIS did reproduce the amplitude of the seasonal cycle in cloud amount. Marchant et al. (2016) note that MODIS cloud top detection becomes less reliable for optically thin clouds at colder temperatures < 240 K, and these associated errors propagate in a cloud phase
identification algorithm. Wilson et al. (2018) compared MODIS cloud single-phase microphysical retrievals (optical depth and effective radius) with retrievals from a surface-based shortwave spectroradiometer on the WAIS, and noticed discrepancies likely related to phase detections errors in low clouds close to the snow surface.

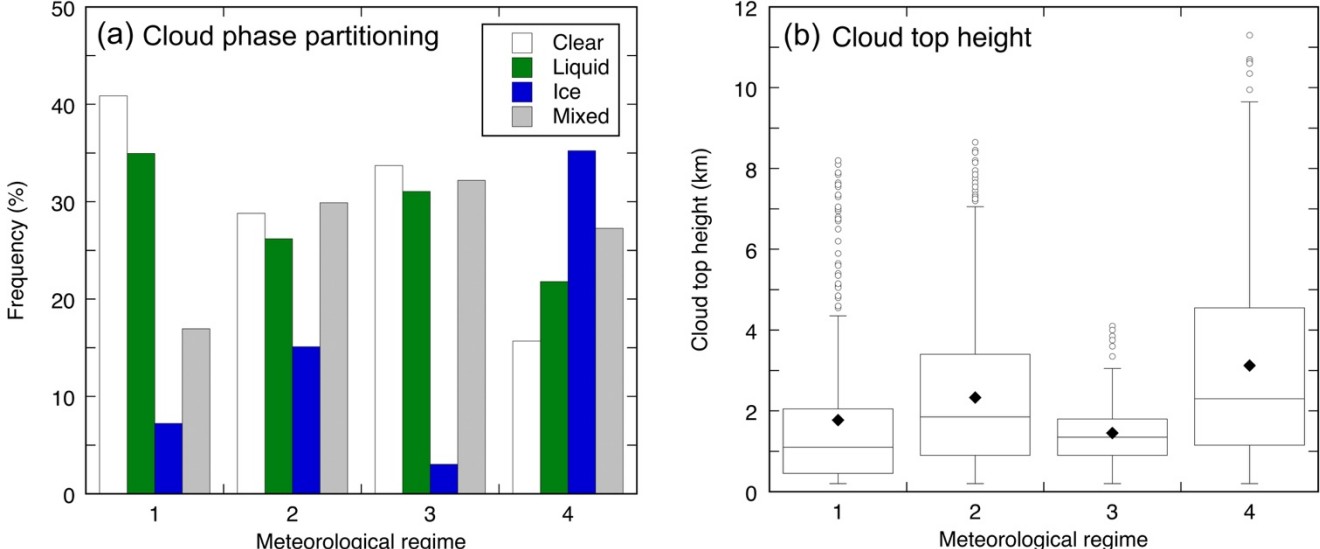

**Figure 4: MODIS retrievals of cloud properties throughout the austral summer 2015-16 in a 1º x 1º latitude x longitude grid cell over McMurdo Station, and sorted by the four meteorological regimes from the k-means clustering analysis: (a) cloud thermodynamic phase; (b) cloud top height. In the second panel the box encloses the middle 50% of the distribution (lower quartile to upper quartile), the median is depicted by the dividing line, and the diamond depicts the mean value. The lines extending above and below each box depict nearly the entire distribution. Individual points above these lines represent individual outliers whose values are greater than the upper quartile value plus 1.5 times the interquartile distance.**

For these reasons, we checked the conclusions in Figure 4 against a surface-based ARM Value Added Product (VAP) from AWARE. We use the THERMOCLDPHASE VAP (de Boer et al., 2011) which contains phase identification in up to ten cloud layers from the High Spectral Resolution Lidar (HSRL). Analogous to Figure 4a, Figure 5 shows the occurrence frequency of clear skies and cloud phase classifications for each of the four regimes, for the highest HSRL-detected cloud layer. There is some qualitative consistency with the MODIS results, particularly for Regime 4, which shows the lowest clear sky frequency and highest ice cloud frequency. Also, the warm Regime 1 shows the greatest occurrence of liquid water cloud. The obvious discrepancy is the severe under-detection of liquid water cloud compared with MODIS, and strong preference for ice cloud in all regimes. This inevitably results from the HSRL's upward-looking view: Silber et al. (2021) analyzed AWARE data from multiple sensors and determined that ~75% of supercooled liquid-bearing clouds precipitate ice particles from the cloud base, and this is consistently detected by the upward-looking lidar while it would almost always be missed by the MODIS satellite retrieval algorithm. Above these same predominantly liquid water clouds, the MODIS cloud phase algorithm would identify most of them as containing mainly liquid water. Despite the discrepancies, both the VAP and

MODIS retrievals show a relative preference for cloud liquid water in the warm regimes and ice water in Regime 4. We now investigate the potential manifestation of these phase contrasts in surface radiometric measurements.

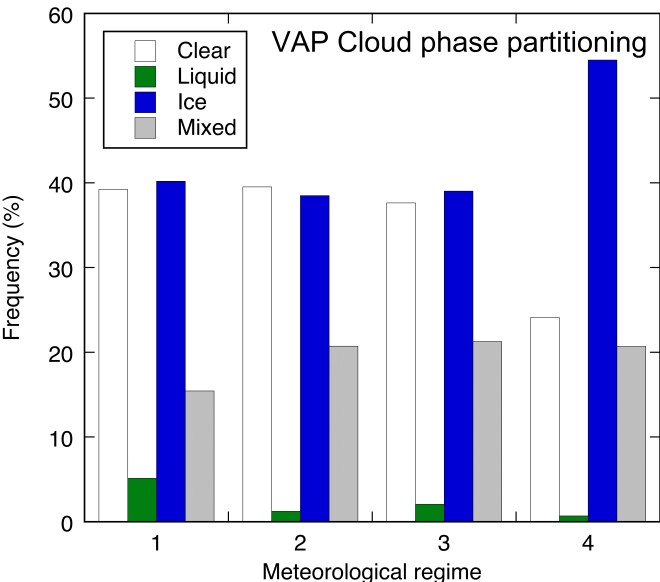

**Figure 5. As in Figure 4a but for the ARM THERMOCLDPHASE High Spectral Resolution Lidar phase retrievals for the highest detected cloud layer above the AWARE site on Ross Island.**

## 3 Results

### 3.1 Cloud Amount and Longwave Irradiance

Figure 6 shows the frequency distribution of TSI-observed cloud coverage for each of the four meteorological regimes. All regimes show 60-70% of sky conditions divided roughly equally between essentially clear skies (TSI cloud amount < 5%) and nearly overcast to overcast (TSI cloud amount > 90%). This is explained by most of the region's cloud coverage being driven by extensive storm systems originating in the Southern Ocean. Comparing Figure 6 with Figure 4a, we notice that MODIS identification of clear skies is 6% larger than TSI in the warmer regimes (1 and 3), which is consistent with a known tendency of whole-sky imagers to underestimate clear sky fraction due to scattering artefacts near the position of the direct solar beam (Pfister et al., 2003; Long et al., 2006). The discrepancies for the warmer regimes are quite small considering the widely differing fields of view between the TSI all-sky camera and the MODIS satellite footprint. In the colder regimes MODIS identification of clear skies is 10-11% smaller than TSI, and these discrepancies for the colder regimes may be analogous to those found in Cossich et al. (2021). Table 2 gives basic statistics for SKYRAD downwelling longwave measurements for each of the four meteorological regimes, for TSI cloud amount < 5% and > 95%. These results are consistent with Regime 1 being the warmest regime and Regime 2 being the coldest. Regime 3, although one of the warmer

**Table 1: Daily occurrences of each *k*-means clustering meteorological regime throughout the austral summer 2015-16.**

|  | December 2015 | January 2016 | February 2016 |
|---|---|---|---|
| Regime 1 | 9, 10, 11, 12, 13, 20, 21, 22, 23, 24, 25, 26, 27 | 6, 7, 8, 9, 10, 11, 12, 13, 14, 15 | none |
| Regime 2 | 1, 2, 3, 8, 18, 19, 28, 29, 30 | 20, 21, 22, 23, 24, 25, 26, 28, 29 | 8, 11, 12, 13, 14, 15, 16, 17, 18, 19, 20, 21 |
| Regime 3 | 14, 15, 16, 17, 31 | 1, 2, 3 | none |
| Regime 4 | 4, 5, 6, 7 | 4, 5, 16, 17, 18, 19, 30, 31 | 1, 2, 3, 4, 5, 6, 7, 9, 10, 22, 23, 24, 25, 26, 27, 28, 29 |

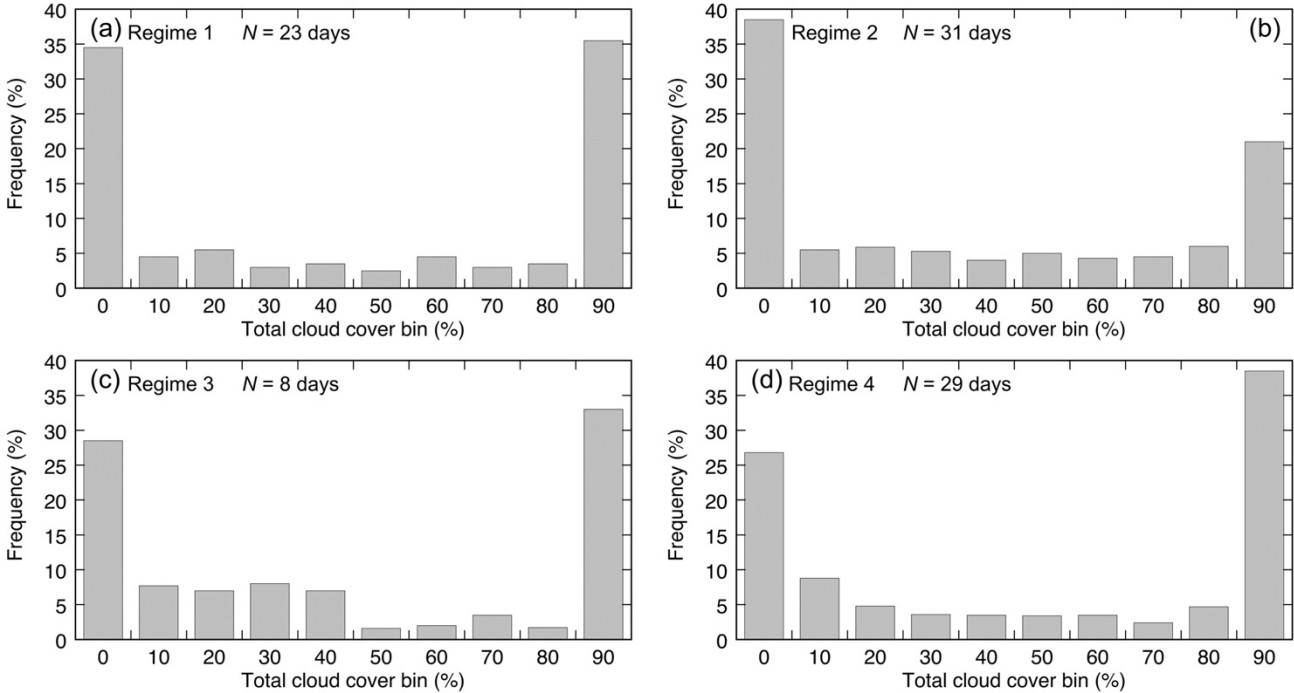

**Figure 6: Histograms of total cloud cover measured by the TSI on Ross Island throughout the austral summer 2015-16, for each of the four meteorological regimes. The bins are identified by their lower bound, with the exception of the clear sky bin, which**
**depicts coverage in the range 0-5%. The number of days each regime occurred is given on each panel.**

regimes, shows relatively low mean downwelling longwave flux under overcast, and this can be attributed to relatively lower cloud optical depths as discussed below.

### 3.2 Shortwave Atmospheric Transmittance

Our objective is to detect contrasts in cloud properties among the four meteorological regimes, and the key diagnostics are
the atmospheric transmittances from the SKYRAD shortwave and MFRSR 870-nm measurements. Here the atmospheric

**Table 2: Statistics for SKYRAD longwave irradiance measurements under each of the four meteorological regimes (W m$^{-2}$), for TSI cloud amounts < 5% (CLR) and > 95% (OVC). The standard deviations are given along with the mean values.**

| Regime | CLR Min | CLR Mean | OVC Min | OVC Mean |
|--------|---------|----------|---------|----------|
| 1 | 184.8 | 203.2 ± 8.2 | 225.8 | 270.0 ± 13.3 |
| 2 | 149.0 | 192.0 ± 20.9 | 197.7 | 260.9 ± 23.1 |
| 3 | 183.7 | 202.6 ± 7.7 | 218.1 | 262.3 ± 18.1 |
| 4 | 161.0 | 191.7 ± 17.9 | 206.8 | 265.8 ± 22.8 |

transmittance is computed as the measured irradiance divided by the product of the extraterrestrial solar irradiance and the cosine of the solar zenith angle, and also correcting for the varying Earth-Sun distance. For the SKYRAD shortwave, we use an extraterrestrial solar irradiance of 1350 W m$^{-2}$, which is typical for the spectral interval covered by pyranometers. For the 870-nm MFRSR channel, we acquired the particular MFRSR Filter 5 spectral response function for the instrument used with AMF2 at AWARE. We used this as a weighting function when summing the MODTRAN (Berk et al., 2014) extraterrestrial solar spectral irradiance over the wavelength interval 851.5-891.5 nm, to derive an extraterrestrial solar irradiance for the 870-nm channel of 0.9376 W m$^{-2}$.

To determine if there are contrasting influences of cloud thermodynamic phase among the meteorological regimes, we examine the SKYRAD shortwave atmospheric transmittance as a function of the 870-nm atmospheric transmittance. The essential physical principle is that the 870-nm irradiance measurement is affected by conservative scattering by cloud particles. At wavelengths < 1100 nm, attenuation of solar radiation by cloud droplets or ice particles is overwhelmingly due to scattering and is nearly entirely independent of wavelength and, in turn, independent of thermodynamic phase or droplet/particle size (e.g., Lubin and Vogelmann, 2011). For measurements that include longer wavelengths, such as SKYRAD shortwave, absorption of radiation by the cloud layer is introduced by the spectral variabilities in the complex refractive indices of liquid and ice water as well as by the difference between the two phases. This spectrally variable absorption can be used to retrieve cloud thermodynamic phase and effective particle size if one has spectrally resolved surface irradiance measurements in the 1.6-micron window (e.g., McBride et al., 2011; Wilson et al., 2018). With respect to the surface energy budget, as measured in the shortwave by the SKYRAD pyranometers, a cloud phase transition from mainly liquid water to substantial ice water content introduces supplemental absorption and attenuation of broadband shortwave surface irradiance for a cloud with the same conservative scattering atmospheric transmittance. This supplemental attenuation occurs at near-infrared wavelengths and is typically in the range 5-10 W m$^{-2}$ for Arctic mixed-phase stratiform clouds (Lubin and Vogelmann, 2011).

Figure 7 shows the SKYRAD shortwave atmospheric transmittance as a function of the 870-nm atmospheric transmittance for each of the four regimes. All of the SKYRAD transmittances lie below the one-to-one correspondence line due to absorption of radiation at near-infrared wavelengths. If this empirical function for one regime lies below that of another

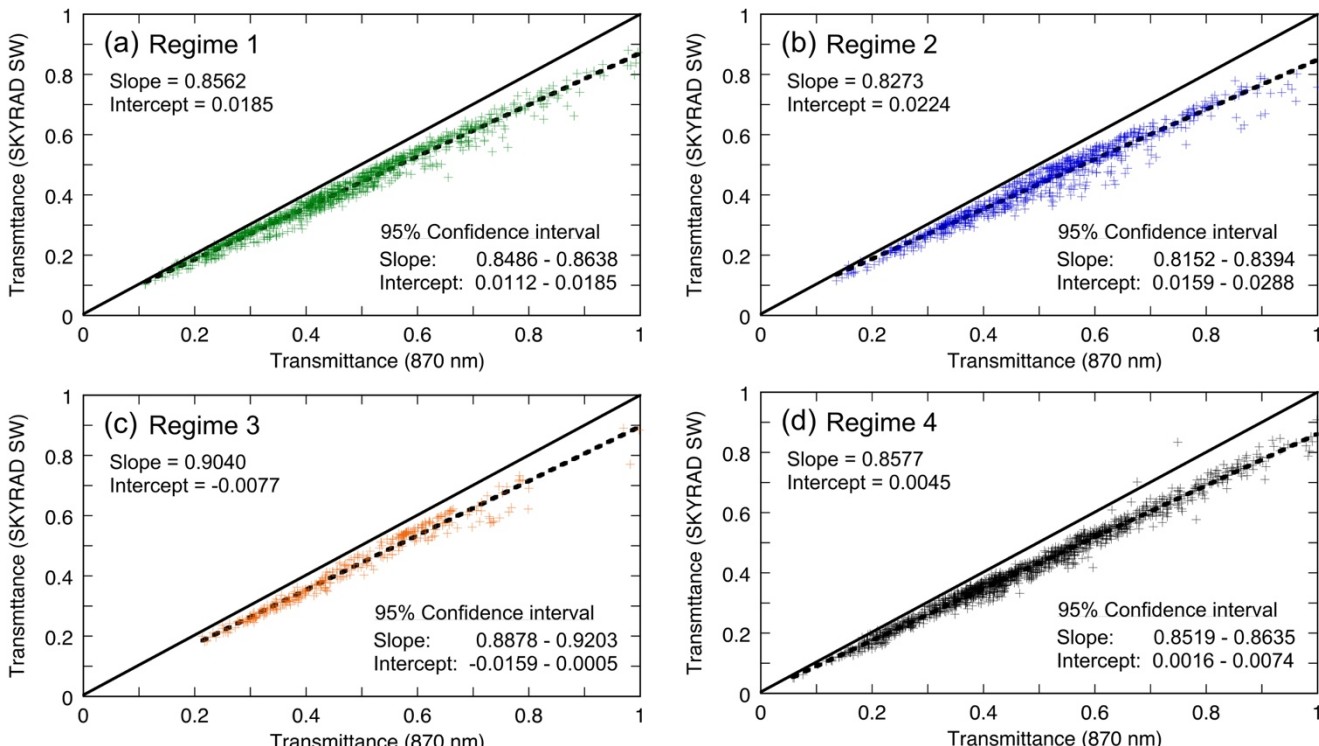

**Figure 7: Broadband shortwave transmittance as measured by the SKYRAD as a function of narrow-band transmittance at 870 nm as measured by the MFRSR under all overcast skies throughout the austral summer 2015-16, for each of the four meteorological regimes. The solid line depicts one-to-one correspondence. The dotted line depicts a linear regression, with the slope and intercept and the 95% confidence intervals on the slope and intercept given in each panel.**

regime, this indicates greater near-infrared absorption. In the cold and dry Antarctic atmosphere, such differences are mainly due to contrasting cloud thermodynamic phase, with the presence of cloud ice water causing more absorption. In Figure 7 we see that the coldest regime (2) has a shallower slope than that for either warm regime (1 or 3), signifying more absorption due to cloud ice, also consistent with Figure 4. Regimes 1 and 4 show essentially the same slope, but Regime 4 shows a smaller intercept and therefore its SKYRAD transmittances on average lie slightly below those of Regime 1, consistent with more cloud ice absorption from the clouds sampled under Regime 4.

### 3.3 Cloud Optical Depth

While the atmospheric transmittance is sufficient to reveal the underlying cloud phase contrasts between meteorological regimes, we can obtain additional information by retrieving the effective cloud optical depth. To retrieve this optical depth we use the discrete-ordinates-based radiative transfer model (Stamnes et al., 1988) adapted for polar clouds by Lubin and Vogelmann (2011). In a plane-parallel model atmosphere based on AWARE sonde data, we iterate on cloud optical depth

until the calculation matches the measured 870-nm surface irradiance. For this we need the 870-nm surface albedo, which is obtained as the ratio of the downward-facing MFR measurement to the upward-facing MFRSR measurement. During summer at the CosRay site the surface had little snow cover and featured mainly bare volcanic rock. Surface albedos evaluated every minute (when the solar elevation was above 10°) ranged from 0.015-0.524, with a seasonal mean value 0.088±0.051. Therefore local multiple reflection of photons between the surface and cloud base plays a minor role in the radiation transport, and the spectral surface irradiance is highly sensitive to cloud optical depth. At the same time we note that this site with its low surface albedo is not completely representative of a snow-covered ice shelf, and some assumptions about snow surface albedo and emissivity are required to generalize these results to other Antarctic locations.

Figure 8 shows the frequency distributions in effective cloud optical depth for each of the four meteorological regimes. These distributions are qualitatively similar to optical depth distributions derived over sea ice by Fitzpatrick and Warren (2005; see for example their Figure 14) and by Fitzpatrick and Warren (2007; see for example their Figure 2). Our optical depth distributions are amenable to an exponential fit, as in these references. However, comparing our distributions with Fitzpatrick and Warren (2007) for the latitude of McMurdo Station, we have overall a shallower decrease in frequency of occurrence with optical depth, more consistent with their result for the latitude range 67.5-70.0°S. This most likely results from our consideration of only summertime data, while their results encompass shipboard measurements from winter, spring and summer seasons.

Figure 9 shows the cumulative frequency distributions in effective cloud optical depth for each of the regimes. Regimes 1 (marine air) and 4 (orographic forcing) tend to have larger optical depths, with ~75% of the values being less than 9 and 8, respectively, and reaching ~95% of the total sample by optical depth values 20 and 19, respectively. Regimes 2 (cold outflow) and 3 (warm but infrequent) tend to have smaller optical depths, with ~75% of the values being less than 5 and 7, respectively, and reaching ~95% of the total sample by optical depth value 13. Although regimes 1 and 4 have similar conservative scattering optical depth distributions, examination of the broadband shortwave SKYRAD observations reveal statistically significant differences in shortwave attenuation for various discrete ranges of optical depth, as described next. One significant result is that for all regimes, between 58-78% of clouds have optical depths ≤ 5. At larger optical depths a cloud emits longwave radiation essentially as a blackbody. For optical depths ≤ 5 the cloud emissivity will vary (including spectrally) as a function of liquid and ice water content content, and cloud particle size. This introduces additional variation in the downwelling longwave radiation, for all four meteorological regimes.

### 3.4 Removing Autocorrelation
The original irradiance measurements and optical depth retrievals are recorded in one-minute intervals. For cloud systems these are generally not independent observations. Significant autocorrelation, if not identified and removed, will artificially

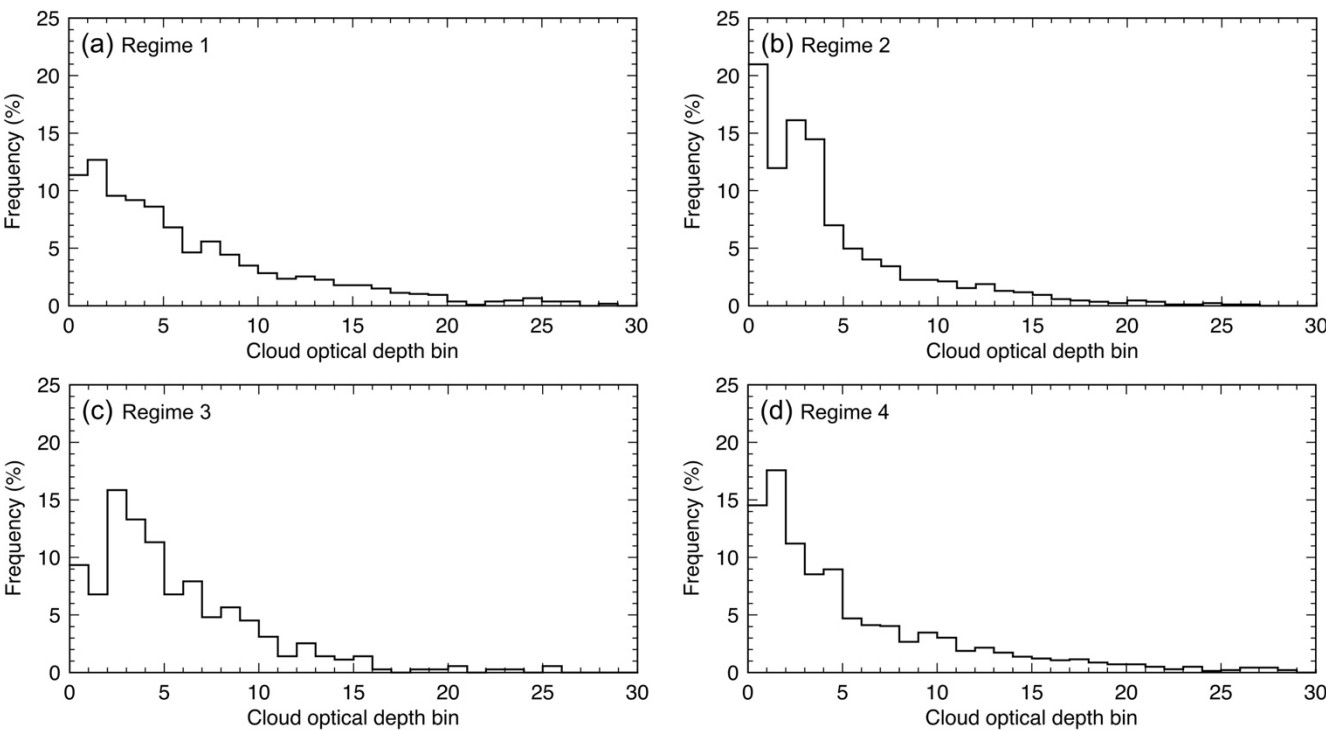

**Figure 8: Frequency distributions of conservative-scattering cloud optical depth retrieved from the 870-nm irradiances under overcast skies, for each of the four meteorological regimes.**

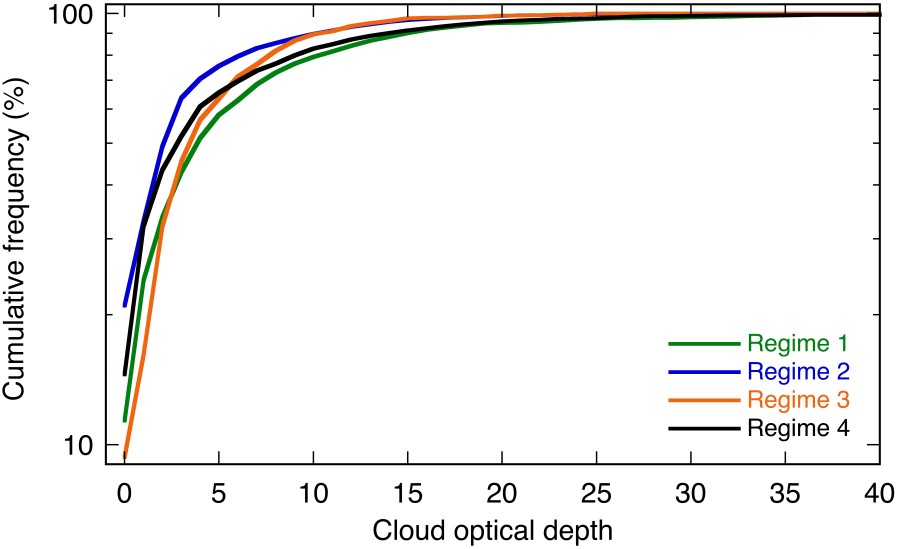

**Figure 9: Cumulative frequency distributions of conservative-scattering cloud optical depth under overcast skies, for each of the four meteorological regimes.**

inflate the statistical significance in any subsequent analysis. We checked for autocorrelation using a time series between 15:00-17:00 UTC on 11 December 2015, during which both the SKYRAD shortwave transmission and the 870-nm cloud optical depth retrievals showed stationary trends. At the original one-minute time interval, both observations showed significant lag-one autocorrelations greater than 0.95. When we average over ten minutes, the lag-one autocorrelation in shortwave transmission drops to 0.12 and the lag-one autocorrelation in optical depth drops to 0.55. Applying the analysis of Bretherton et al. (1999), these reduced autocorrelations signify that we can use the ten-minute averages for statistical analysis if we further reduce the effective sample size to $N_{eff} = 0.88N$.

### 3.5 Contrasts Among the Meteorological Regimes

In Figure 10 we compare the SKYRAD shortwave atmospheric transmission observations for meteorological regimes 1, 2 and 4. We do not consider the third cluster as it occurs much less frequently (most of its observations span a four-day period 13 December 2015 through 3 January 2016). In this figure we have sorted the observations into conservative scattering optical depth bins 0-2, 2-4, 4-6, 6-8, 8-10, 10-15, and 15-20. We omit the larger optical depths, which occur infrequently. For each optical depth bin we plot the mean and standard deviation of the corresponding SKYRAD shortwave transmissions. To emphasize the comparisons between pairs of meteorological regimes, a separate panel shows the transmission difference for each optical depth bin, along with the statistic from Student's $t$-test.

Comparing regimes 1 and 4 (Figure 10a,b), we see that shortwave surface irradiance is consistently smaller under the orographic regime's ice and mixed-phase clouds, for all optical depth bins. This difference is statistically significant in all but the smallest optical depth bin. Comparing regimes 2 and 4 (Figure 10c,d), shortwave transmission in the orographic regime is noticeably smaller than in the cold regime for five of the seven optical depth bins, and four of these bins show a statistically significant difference. The cold regime shows lower shortwave transmission than the orographic regime under moderate optical depths 6-10. Comparing regimes 1 and 2 (Figure 10e,f), the cold regime shows lower shortwave irradiance for cloud optical depths 6 and larger, with statistical significance for the optical depth range 6-10. For smaller optical depths the maritime regime has statistically significant lower shortwave irradiance.

We tested whether these results could be caused by differences per optical depth bin between the paired clusters. We evaluated the mean optical depths in all of the bins and found that they are all very similar, even in the largest two bins. In the largest bin (15-20) the mean optical depths differ by less than 0.4, and the differences are much smaller for all the other bins. Therefore the shortwave irradiance differences described by Figure 10 are not artifacts of contrasting optical depth distributions in the bins.

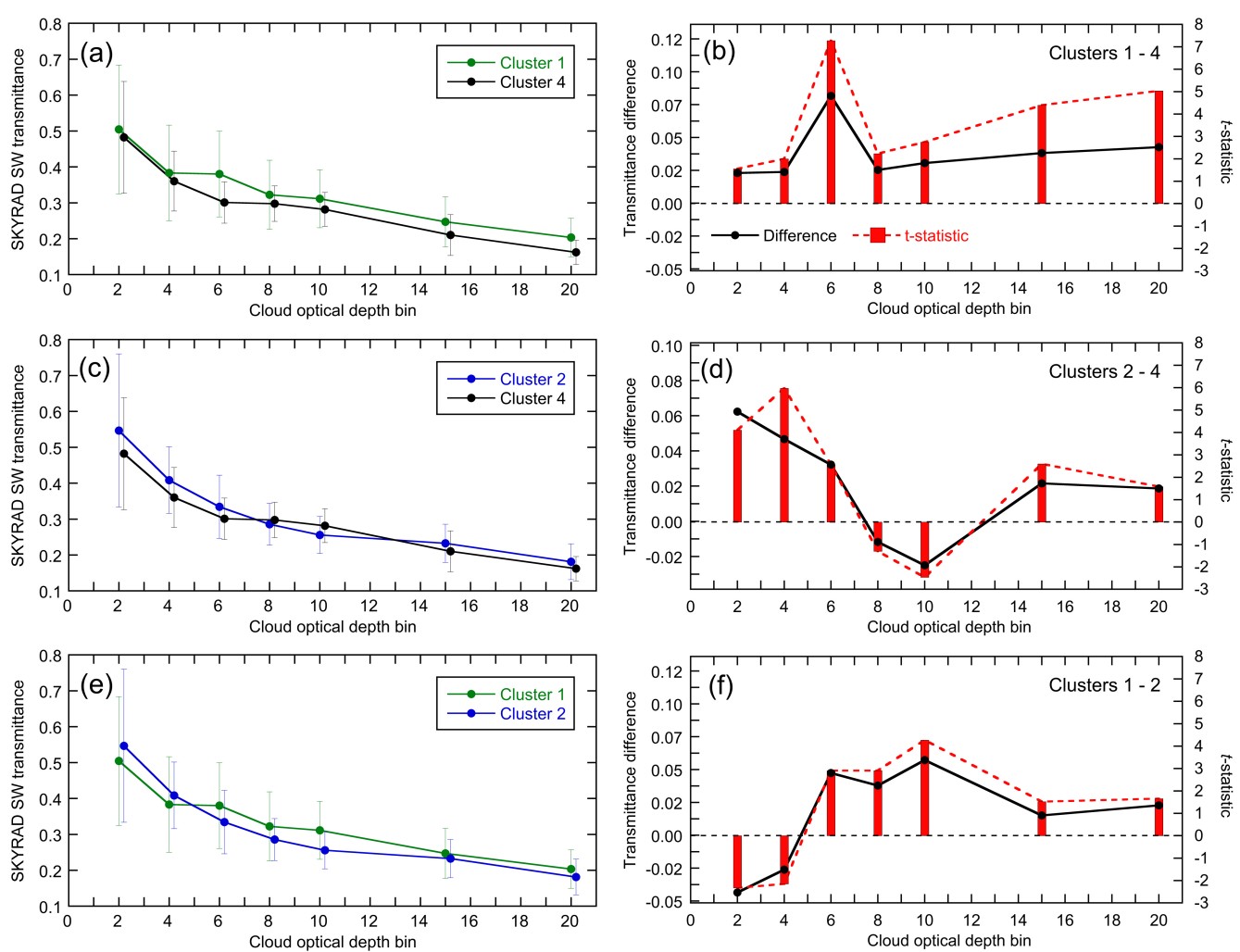

**Figure 10: SKYRAD measured shortwave atmospheric transmittance as a function of conservative-scattering cloud optical depth bin, comparing meteorological regimes 1, 2 and 4. Each pair of panels shows the individual mean transmittances for the optical depth bins (left), then the transmittance difference between the two compared regimes (in legend) along with the statistic from Student's *t*-test (right). Error bars are plus/minus one standard deviation. Cloud optical depths are of width 2 for the first five bins, then 5 for the largest bins, and are identified on these plots by their upper bound. For the range of sample sizes considered here, a *t*-statistic ≥ 2.0 indicates statistical significance.**

Table 3 summarizes the mean radiative flux differences between three of the regimes. For the longwave we can use the mean values from Table 2 directly. The shortwave has diurnal solar zenith angle dependence, and to compute a representative seasonal value for each regime we binned the observations by optical depth as described above, and then by solar zenith angle in five-degree bins between 55º and 80º. The representative seasonal value is evaluated as the sum over all the optical depth and solar zenith angle bins, weighted by the observation's frequency of occurrence in each bin. As these observations

**Table 3: Mean surface radiative flux differences between three pairs of meteorological regimes (W m$^{-2}$). The downwelling values are derived from the observations as described in Figure 10 and the text. The net radiative flux is an estimate extrapolating the Ross Island observations to an ice sheet using representative broadband snow albedo and emissivity values.**

| | Observed at Ross Island | | Extrapolated to Ice Sheet |
| --- | --- | --- | --- |
| Regime Comparison | Downwelling Shortwave | Downwelling Longwave | Net Radiative Flux |
| 1 - 4 | 6.6 | 4.2 | 5.9 |
| 2 - 4 | 10.0 | -4.9 | 8.7 |
| 1 - 2 | -2.6 | 9.1 | -2.2 |

from the CosRay site are not entirely representative of a snow-covered ice sheet or ice shelf, we can make a first order estimate for a snow surface from these observations using representative values of 0.88 for broadband snow albedo under cloud cover (Lubin et al., 2023) and 0.99 for broadband snow emissivity (e.g., Warren, 1982). Table 3 shows that overall, the shortwave differences are more significant than the longwave differences, although both are detectable with statistical significance.

Regarding the longwave irradiance, we can also make a regime comparison binned by cloud optical depth as shown in Figure 11. This figure reveals the contrasting influences of optically thin versus optically thick clouds to the longwave irradiance differences listed in Table 3. Comparing Regimes 1 and 4, the positive longwave irradiance difference is due primarily to clouds with optical depths between 5-10 (radiating effectively as blackbodies), with a contribution from the optically thinnest clouds. Here then the longwave irradiance difference between these regimes is due mainly to differences in average cloud base temperature. Comparing Regimes 2 and 4, the negative longwave irradiance difference is due mainly to optically thin clouds, and therefore involves microphysical differences as well as cloud effective temperature. Comparing Regimes 1 and 2, per Figure 9 the warmer regime has larger cloud optical depth overall, and this accounts for the relatively large positive longwave flux difference between these regimes.

## 4 Conclusion

The contrast between the predominantly liquid water content of the low clouds in the onshore flow regime with marine air mass origin (Regime 1), versus the overwhelming ice water and mixed-phase content of the geometrically extensive clouds in the orographic southerly-flow regime (Regime 4), is consistent with the statistically significant differences in cloud attenuation of shortwave irradiance at the Antarctic surface for nearly all conservative scattering cloud optical depth. Generally speaking, when applying this analysis involving near-infrared atmospheric absorption, one should account for influences of changing water vapor across meteorological regimes. However, in the very cold and dry Antarctic atmosphere considered here, the fact that the colder regimes (2 and 4) show greater shortwave attenuation that the warm regime signifies that these differences are not due to water vapor but to contrasts in cloud ice water content.

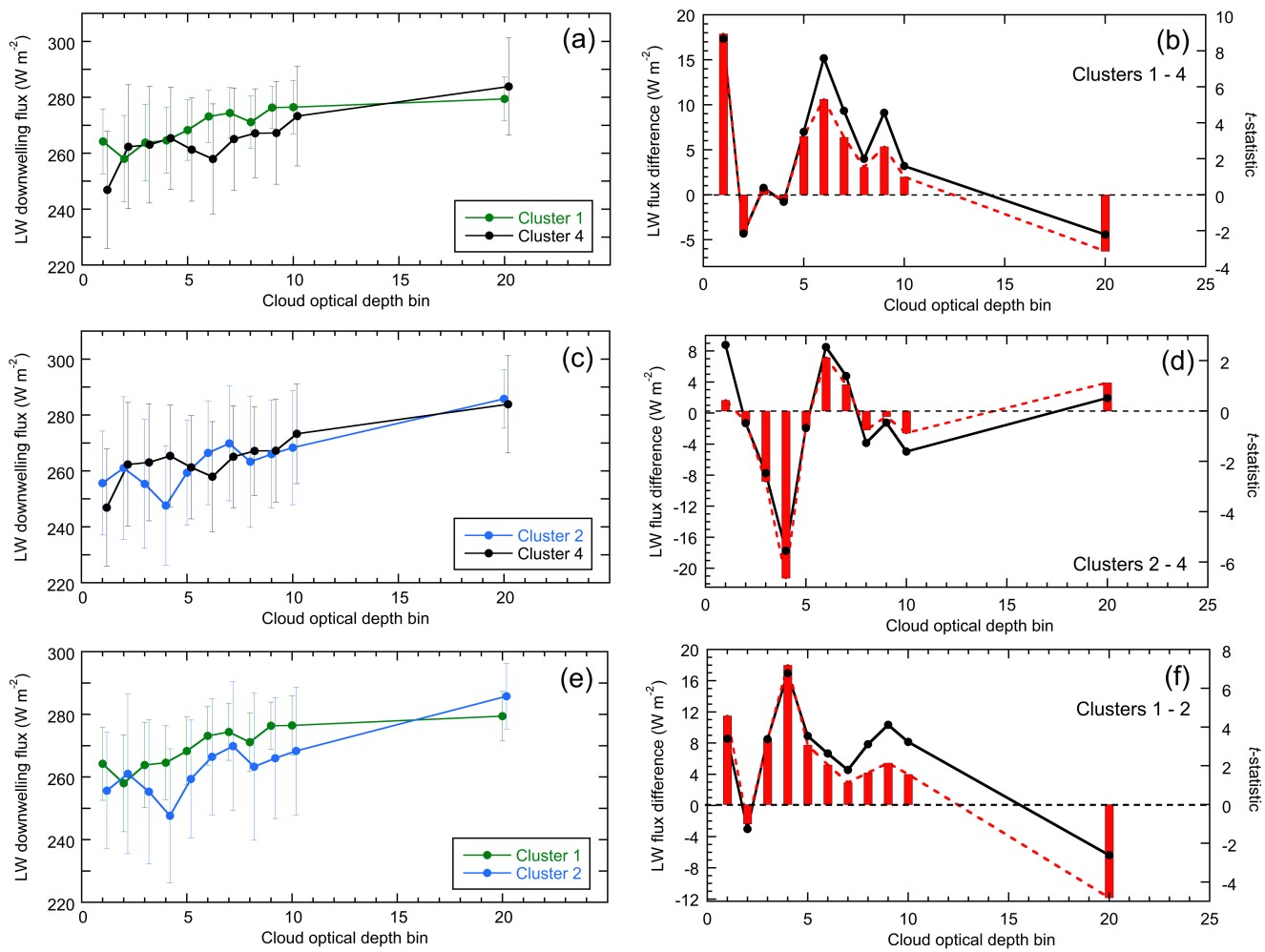

**Figure 11: SKYRAD measured longwave irradiance (flux) as a function of conservative-scattering cloud optical depth bin, comparing meteorological regimes 1, 2 and 4. Each pair of panels shows the individual mean irradiances for the optical depth bins (left), then the irradiance difference between the two compared regimes (in legend) along with the statistic from Student's *t*-test (right). Error bars are plus/minus one standard deviation. Cloud optical depths are of width 1 for the first ten bins and are**

455 **identified by their upper bound, followed by a single bin for the optical depth range 10-20.**

We note that the goal of detecting net surface radiation balance changes of order 10 W m$^{-2}$ is roughly the same order as the calibration uncertainty with pyranometers and pyrgeometers. However, one important consideration is that with overcast skies, under which the upwelling and downwelling radiation is generally diffuse, measurement precision with these

460 instruments is generally very stable. Under partly cloudy or clear skies, where the direct solar beam is present, temperature differentials within the instruments can generate radiometric offsets in the range 10-15 W m$^{-2}$ (Bush et al., 2000). Therefore to associate measured differences of order 10 W m$^{-2}$ with changes in ice melt or retention, one must have either (1) a well-

calibrated time series of pyranomter and pyrgeometer measurements conclusively and independently associated with a specific surface melt event, as in Nicolas et al. (2017), or (2) a large enough data sample that the measured differences become statistically significant. In this work we just barely meet the second criteria with three summer months of data sorted into four meteorological regimes. This study therefore indicates the need for long duration radiometer deployment in the Antarctic to acquire datasets relevant to cryospheric change, and this in turn implies requirements for instrument maintenance and cleanliness that cannot be taken for granted in the deep field.

In support of these conclusions, Scott and Lubin (2016) identified a greater vertical extent of cloud ice water in Antarctic clouds as a marked contrast with the high Arctic. Additionally, Scott et al. (2017) show that orographically driven ice and mixed-phase clouds occur frequently and extensively not just in the vicinity of the Transantarctic Mountains but also in various climate-change-sensitive regions such as the northern extremity of the RIS, Ellsworth Land and the Ronne-Filchner Ice Shelf. In studies of either summertime surface melting or longer-term ice mass balance, in which the SEB needs detailed evaluation, the impact of cloud phase on the radiative components should be accounted for and can be measured by the types of instruments used in this study.

Three years before AWARE, Scott and Lubin (2014) operated a shortwave spectroradiometer at Ross Island between October 2012 and February 2013, and analyzed the evolution of surface shortwave spectral irradiance in five case studies of several days each. One of these case studies involved an onshore flow consistent with the first meteorological regime described here, and these low marine stratiform clouds showed a wide range in optical depth and near-infrared attenuation mainly consistent with liquid water. Two other case studies involved cloud systems that were influenced by the Transantarctic Mountains, and these clouds showed greater vertical extent and near-infrared attenuation consistent with the excess absorption of cloud ice water relative to the liquid phase. This earlier study used measurements of spectral shortwave irradiance which to-date are not commonly available except at well-instrumented sites such as the ARM User Facility (Riihimaki et al., 2021).

However, the low-cost, low-power, rugged and readily transportable instruments used in this study can easily be configured to accompany deep-field glaciological or geological expeditions, and this concept was demonstrated recently at Siple Dome in West Antarctica (Lubin et al., 2023). There are naturally some limitations if these radiometers are left unattended, in particular, ice accretion and riming on the optics (e.g., Cox et al. 2023). If the radiometers accompany an expedition then presumably a team member can ensure optical cleanliness on a daily basis. Also, if there is sufficient power to ventilate the radiometers with blowers this will prevent most optical fouling except in the harshest conditions. At the same time, unattended shortwave and longwave radiometers have been successfully deployed with Antarctic automatic weather stations and have yielded long time series with useful results when relative values are analyzed or suitable parameterizations are developed to account for periods of icing (van den Broeke et al., 2004).

Additionally, some Antarctic research stations have existing atmospheric observation capabilities yielding important climatological results. For example, at Princess Elizabeth Station in East Antarctica, meteorological, ceilometer and precipitation measurements have revealed the influence of atmospheric rivers on accumulation and ice mass balance (Gorodetskaya et al., 2014; 2015). The British Antarctic Survey has conducted aircraft in situ cloud microphysical measurements based out of Rothera Station, which have revealed the importance of secondary ice production at temperatures just below freezing (Lachlan-Cope et al., 2016). Augmenting programs such as these with long-term filter and broadband radiometer observations might further enhance their value. This study demonstrates that extremely remote regions on Earth don't necessarily need to wait for the most advanced atmospheric instrumentation to see advances in understanding, and that careful use of basic radiometric instruments in conjunction with satellite remote sensing and meteorological reanalysis can yield useful results pertaining to a region's climatological questions.

**Data Availability**

All AWARE data are archived with the US Department of Energy Atmospheric Radiation Measurement (ARM) program (www.arm.gov). MODIS cloud satellite data products are obtained from the NASA Level-1- and Atmosphere Archive & Distribution System (LAADS) Distributed Active Archive Center (DAAC). ERA-Interim data were originally obtained from a downloadable archive at the European Centre for Medium-Range Weather Forecasts (ECMWF), and can still be obtained from ECMWF by special request. The download service has recently been discontinued for ERA-Interim as it has been superseded by ERA5 reanalysis which is publicly available at the European Copernicus Climate Change Service (C3S) Climate Data Store. The AWS data are available at the University of Wisconsin Antarctic Meteorological Research Center (https://amrc.ssec.wisc.edu).

**Author Contribution**

KS performed the data analysis and interpretation as part of his Master of Science thesis at the Scripps Institution of Oceanography. RS provided the $k$-means clustering analysis for identifying the meteorological regimes. MLG provided the satellite data and interpretation. AV advised on radiative transfer analysis and manuscript preparation. DL served as thesis advisor for KS and prepared the manuscript for publication.

**Competing Interests**

The authors declare that they have no conflict of interest.

**Acknowledgments**

This work was supported by the US National Science Foundation under PLR-1443549 and by the US Department of Energy under DE-SC0017981 and DE-SC0021974. RCS was supported by NASA Earth & Space Science Fellowship award NNX15AN45H. AMV is supported by the US Department of Energy under contract DE-SC0012704. The AWS program is supported by the US National Science Foundation under ANT-1543305.

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
