# Peer review of "Broadband and filter radiometers at Ross Island, Antarctica: Detection of cloud ice phase versus liquid water influences on shortwave and longwave radiation"

_EGUsphere, 2023_

## Author Comment (AC1)

**Manuscript egusphere-2023-1665 – Replies to Reviewers**

The reviewers have made some valuable suggestions and corrections to significantly improve the quality of the manuscript. All of these suggestions are straightforward to implement, and we look forward to preparing a stronger paper in response to these reviews.

Replies to Reviewer #1:

*This paper uses basic radiometric measurements to calculate cloud optical depths and cloud transmissions and infer the presence of ice or mixed phase clouds. The cloud optical depth calculated using wavelength 870 is not affected by ice clouds whereas broadband shortwave transmission is because of ice bands in the near-infrared. In other words cloud radars, lidars, and other sophisticated equipment are not necessary to obtain fundamentally useful data that can distinguish ice and mixed phase clouds from water clouds.*

*I think the authors demonstrate their point that fairly fundamental radiation measurements can be used at sites in the Antarctic. However, I am not sure that the measurements that they suggest can detect the 10 W/m^2 changes in the net surface radiation (lines 56-58) that could induce ice melt or retention.*

We are grateful that the reviewer endorses our conclusion that these kinds of radiation measurements can be made in the Antarctic environment. The revised manuscript will clarify the context about application to studies of ice melt or retention, and will further discuss the sampling requirements to reach conclusions involving changes of order 10 W/m^2.

*Line 70: indicates about six weeks of measurements; is that correct?*

Yes, that is correct.

*Line 150: "... one describes with a unique large-scale circulation pattern." Is this correctly stated?*

This sentence will be clarified in the revision.

*A map with labels of the geographic sites discussed in these paragraphs would help for those not familiar with Antarctic geography.*

The revision will provide a map as requested, for a new "Figure 1."

Reviewer #2:

*Review of the research article entitled "Broadband and filter radiometers at Ross Island, Antarctica: Detection of cloud ice phase versus liquid water influences on shortwave and longwave radiation" by Kristopher Scarci et al.. (MS No.: egusphere-2023-1665)*

*The authors present a study examining the impact of cloud phase and properties on the surface energy budget based on ground-based radiometric instruments, supported by reanalysis and satellite observations. Their primary objective is to establish a connection between flux measurements and cloud properties using basic instruments deployable in remote locations. This holds particular relevance in Antarctica, where even small changes in the net surface budget could initiate melting processes.*

*The study is well-written, carefully argued and gives important insights into the role of the meteorological patterns and cloud properties on the radiative budget. However, their method is applied to a limited dataset without really enlightening our comprehension of the Antarctica atmospheric system and surface energy balance. As it stands, the research appears more as a proof of concept, aligning more closely with a technical journal such as AMT. To align with the scope and standards of ACP, the study requires a more in-depth analysis to better integrate with the current state-of-the-art research and contribute to a broader understanding of how cloud microphysical properties influence the surface energy budget. Before publication in ACP, several key points need addressing to elevate the study to meet the journal's standards.*

We thank the reviewer for these several suggestions to improve the manuscript, and we are grateful that the reviewer finds the paper to be well-written and carefully argued. We note that we first submitted this manuscript to AMT, whose editor recommended that it would be more suitable for ACP.

***Major comments:***

*1. Scientific significance for the ACP community*

*The authors leverage data from the AWARE campaign (2015-2016) to scrutinize the impact of atmospheric regimes and cloud properties on the surface radiative budget. However, the study encounters limitations arising from the relatively sparse number of samples, both spatially and temporally (e.g., regime 3 was observed on only 8 days), prompting questions about its representativeness or applicability to Antarctica's meteorological patterns. Notably, the AWARE campaign coincided with a period of high global-average temperatures during an El Niño event, further raising considerations about the broader context of the study.*

*To enhance the paper's scope, it is imperative to contextualize the analyses and conclusions within the current state-of-the-art research. This approach not only amplifies the scientific significance but also extends the benefits to a broader community. In this regard, I make a couple of suggestions, non exhaustive, that could help enhancing the generalizability of the conclusions:*

> *• The identification of the meteorological regimes could be put in the context of the paper from Scott et al., 2019. How the four clusters identified in this paper relate to the nine they found, and to the trends they identified.*

We can provide this context as suggested, (a) with reference to Scott et al. 2019 (the k-means clustering method is essentially the same, and (b) earlier work specific to McMurdo (Scott & Lubin 2014, JGR; and Silber et al. 2019, JGR).

> • *The cloud properties retrieved in this study could be compared with the literature you cited. For example, the histogram of cloud optical depth (Figure 5) could be compared with the exponential fit of Fitzpatrick and Warren (2005).*

Fitzpatrick and Warren (2005), and two other papers in that series, are indeed excellent studies for comparison with this work, and will provide these comparisons in the revised manuscript.

*2. Up-to-date reanalysis data set*

*ERA-5 replaced ERA-Interim in 2019, with improved vertical and spatial resolutions, and a newer Integrated Forecasting System. The ECMWF reanalyses are still experiencing warm biases for 2 m air temperature in polar regions, when compared with ground-based observations, with significant differences between ERA-Interim and ERA5 (see, for example, Jonassen et al., 2019; King et al., 2022; Zhu et al., 2021; Wang et al., 2019).*

*In the manuscript, you wrote "For the purposes of this work ERA5 reanalyses are essentially identical to ERA-Interim.", but I would like to understand what the expected implications are on the identification of the meteorological regimes. I understand that updating the reanalysis data set would require a lot of work, but could you justify why it is not necessary, and if it is supported by the AWARE observations.*

This is a very good point regarding use of an older generation of reanalysis data. We will address it by providing new figures comparing the specific meteorological variables used for the k-means clustering in both ERA5 and ERA-Interim. The differences are small as shown in the following preliminary sketches:

[Figure]

*3. Geographical components*

*The paper is missing a map locating the ARM sites and the regions mentioned in the manuscript. It is important for the readers that are not very familiar with those regions, to better understand the links with the meteorological regimes. See for example the figure 1 in Scott et al., 2019, or the figure 1 in Silber et al., 2019.*

*Also, the Figure 1 shall contain latitudes and longitudes.*

As also requested by Referee #1, a map with these details will be provided.

*4. Relevance of the MODIS cloud products for this study*

*In this paper, MODIS is used to assess the cloud phase and cloud top height over McMurdo station during the summer 2015-2016. As mentioned in the manuscript, some improvements have been made in polar regions (Frey, 2008), but inconsistencies still exist for cloud occurrence (e.g., Cossich et al., 2021; Marchant et al., 2016) and cloud properties (e.g., Wilson et al., 2018).*

*The impact of uncertainties in MODIS products on the analysis is an important consideration, especially in the presence of multi-layer clouds or potential omissions of certain clouds. The authors should explicitly address how these uncertainties might affect their findings, particularly in scenarios where the MODIS data might misrepresent the actual cloud conditions.*

The revised manuscript will include more extensive discussion about MODIS cloud retrieval uncertainties at high latitudes and how they bear on this work's conclusions.

*My other comment is why the authors didn't use ground-based observations (radar, lidar, ARM VAP, ...) acquired during AWARE to do their analysis or at least assess the cloud properties retrieved by MODIS (see Minor comment below on Figures 2 and 3).*

The revised manuscript will use at least two of the ARM Value Added Products (VAPs) to assess the cloud phase retrieved by MODIS, and also to independently verify our conclusions related to cloud phase. Such useful VAPs include the Active Remote Sensing of Cloud Locations (ARSCL) and the THERMOCLDPHASE.

***Minor comments***:

*Page 4, Line 123: In the past, maintaining the measurement quality of some radiometers has been a challenge when used in polar environment, for example when icing appears on the optics (Cox et al., 2021). Could you comment on how feasible it would be to have the suite of instruments you suggest in unattended remote location?*

There have been some broadband radiometers occasionally deployed with Antarctic Automatic Weather Stations, and similar net radiation measurements have been made in East Antarctica.

These will be reviewed in the revised manuscript, with emphasis on the potential or limitations involved with unattended operations.

*Page 8, Table 1: The date January 26, 2016, appears 2 times.*

The table will be corrected.

*Pages 8 and 9, Figure 2a and 3: Could you comment on the differences between MODIS clear sky and TSI cloud cover. For example, MODIS identified clear sky during 15%, while TSI observed clear sky for more than 25 %.*

This is ultimately related to differences in field of view between space-based and surface-based sensors, and this will be discussed in the revision.

*Page 12, Figure 6: The figure may be easier to read using a log-scale.*

This change will be made.

*Page 13, Line 306: Particle size could be another important parameter influencing the emissivity of optically thin clouds.*

This will be mentioned in the revision.

*Pages 14 and 16: Figures 7 (b, d, f) and 8 (b, d, f): It would be easier to read with the dashed line crossing 0 on both y-axes, and a legend for the red bars and black line.*

These changes will be made in the revision.

---

## Author Response (AR1)

**Manuscript egusphere-2023-1665 – Replies to Reviewers with Revisions**

The reviewers have made some valuable suggestions and corrections to significantly improve the quality of the manuscript. All of these suggestions are straightforward to implement, and we look forward to preparing a stronger paper in response to these reviews.

Note that when we specify "new lines" in these replies, they refer to lines in the Track Changes version of the revision, which has some slight formatting differences from the version of the revision with changes accepted.

**Replies to Reviewer #1:**

*This paper uses basic radiometric measurements to calculate cloud optical depths and cloud transmissions and infer the presence of ice or mixed phase clouds. The cloud optical depth calculated using wavelength 870 is not affected by ice clouds whereas broadband shortwave transmission is because of ice bands in the near-infrared. In other words cloud radars, lidars, and other sophisticated equipment are not necessary to obtain fundamentally useful data that can distinguish ice and mixed phase clouds from water clouds.*

*I think the authors demonstrate their point that fairly fundamental radiation measurements can be used at sites in the Antarctic. However, I am not sure that the measurements that they suggest can detect the 10 W/m^2 changes in the net surface radiation (lines 56-58) that could induce ice melt or retention.*

We are grateful that the reviewer endorses our conclusion that these kinds of radiation measurements can be made in the Antarctic environment. We have added the following text in the Conclusions:

We note that the goal of detecting net surface radiation balance changes of order 10 W m$^{-2}$ is roughly the same order as the calibration uncertainty with pyranometers and pyrgeometers. However, one important consideration is that with overcast skies, under which the upwelling and downwelling radiation is generally diffuse, measurement precision with these instruments is generally very stable. Under partly cloudy or clear skies, where the direct solar beam is present, temperature differentials within the instruments can generate radiometric offsets in the range 10-15 W m$^{-2}$ (Bush et al., 2000). Therefore to associate measured differences of order 10 W m$^{-2}$ with changes in ice melt or retention, one must have either (1) a well-calibrated time series of pyranomter and pyrgeometer measurements conclusively and independently associated with a specific surface melt event, as in Nicolas et al. (2017), or (2) a large enough data sample that the measured differences become statistically significant. In this work we just barely meet the second criteria with three summer months of data sorted into four meteorological regimes. This study therefore indicates the need for long duration radiometer deployment in the Antarctic to acquire datasets relevant to cryospheric change, and this in turn implies requirements for instrument maintenance and cleanliness that cannot be taken for granted in the deep field.

*Line 70: indicates about six weeks of measurements; is that correct?*

Actually three months of measurements. This has been clarified in new lines 73-75.

*Line 150: "... one describes with a unique large-scale circulation pattern." Is this correctly stated?*

This sentence has been clarified in new lines 156-157.

*A map with labels of the geographic sites discussed in these paragraphs would help for those not familiar with Antarctic geography.*

This map has been included as the new Figure 1.

**Replies to Reviewer #2:**

*Review of the research article entitled "Broadband and filter radiometers at Ross Island, Antarctica: Detection of cloud ice phase versus liquid water influences on shortwave and longwave radiation" by Kristopher Scarci et al.. (MS No.: egusphere-2023-1665)*

*The authors present a study examining the impact of cloud phase and properties on the surface energy budget based on ground-based radiometric instruments, supported by reanalysis and satellite observations. Their primary objective is to establish a connection between flux measurements and cloud properties using basic instruments deployable in remote locations. This holds particular relevance in Antarctica, where even small changes in the net surface budget could initiate melting processes.*

*The study is well-written, carefully argued and gives important insights into the role of the meteorological patterns and cloud properties on the radiative budget. However, their method is applied to a limited dataset without really enlightening our comprehension of the Antarctica atmospheric system and surface energy balance. As it stands, the research appears more as a proof of concept, aligning more closely with a technical journal such as AMT. To align with the scope and standards of ACP, the study requires a more in-depth analysis to better integrate with the current state-of-the-art research and contribute to a broader understanding of how cloud microphysical properties influence the surface energy budget. Before publication in ACP, several key points need addressing to elevate the study to meet the journal's standards.*

We thank the reviewer for these several suggestions to improve the manuscript, and we are grateful that the reviewer finds the paper to be well-written and carefully argued. We note that we first submitted this manuscript to AMT, whose editor recommended that it would be more suitable for ACP.

*Major comments:*

*1. Scientific significance for the ACP community*

*The authors leverage data from the AWARE campaign (2015-2016) to scrutinize the impact of atmospheric regimes and cloud properties on the surface radiative budget. However, the study encounters limitations arising from the relatively sparse number of samples, both spatially and temporally (e.g., regime 3 was observed on only 8 days), prompting questions about its representativeness or applicability to Antarctica's meteorological patterns. Notably, the AWARE campaign coincided with a period of high global-average temperatures during an El Niño event, further raising considerations about the broader context of the study.*

*To enhance the paper's scope, it is imperative to contextualize the analyses and conclusions within the current state-of-the-art research. This approach not only amplifies the scientific significance but also extends the benefits to a broader community. In this regard, I make a couple of suggestions, non exhaustive, that could help enhancing the generalizability of the conclusions:*

> *• The identification of the meteorological regimes could be put in the context of the paper from Scott et al., 2019. How the four clusters identified in this paper relate to the nine they found, and to the trends they identified.*

We have added the following text in new lines 206-215:

Scott et al. (2019) applied the *k*-means clustering technique to ERA-Interim data over a larger domain to derive nine synoptic patterns influencing surface melt in West Antarctica. Our Regime 1 here is analogous to Pattern 7 of Scott et al. (2019), which favors surface melt. Our cold Regime 2 here is similar to Pattern 2 of Scott et al. (2019). Our Regime 3, which as a small impact on temperature anomalies on the RIS but features warming and melting on the Ronne-Filchner Ice Shelves, is similar to Pattern 5 of Scott et al. (2019). Our unique Regime 4 has similarities with Patter 3 of Scott et al. (2019), particularly regarding warm temperature anomalies over the southern RIS, although in this analysis the driving cyclone appears stronger and situated slightly west of the cyclone derived in Scott et al. (2019). Similarly, Silber et al. (2019) applied a self-organizing map (SOM) technique to classify synoptic regimes influencing Ross Island during AWARE. Our Regimes 1-4 are similar to the Silber et al. (2019) SOM numbers 4, 3, 7 and 6, respectively.

> *• The cloud properties retrieved in this study could be compared with the literature you cited. For example, the histogram of cloud optical depth (Figure 5) could be compared with the exponential fit of Fitzpatrick and Warren (2005).*

We have added the following text in new lines 424-430:

These distributions are qualitatively similar to optical depth distributions derived over sea ice by Fitzpatrick and Warren (2005; see for example their Figure 14) and by Fitzpatrick and Warren (2007; see for example their Figure 2). Our optical depth distributions are amenable to an exponential fit, as in these references. However, comparing our distributions with Fitzpatrick and Warren (2007) for the latitude of McMurdo Station, we have overall a shallower decrease in frequency of occurrence with optical depth, more consistent with their result for the latitude

range 67.5-70.0°S. This most likely results from our consideration of only summertime data, while their results encompass shipboard measurements from winter, spring and summer seasons.

*2. Up-to-date reanalysis data set*

**ERA-5 replaced ERA-Interim in 2019, with improved vertical and spatial resolutions, and a newer Integrated Forecasting System. The ECMWF reanalyses are still experiencing warm biases for 2 m air temperature in polar regions, when compared with ground-based observations, with significant differences between ERA-Interim and ERA5 (see, for example, Jonassen et al., 2019; King et al., 2022; Zhu et al., 2021; Wang et al., 2019).**

**In the manuscript, you wrote "For the purposes of this work ERA5 reanalyses are essentially identical to ERA-Interim.", but I would like to understand what the expected implications are on the identification of the meteorological regimes. I understand that updating the reanalysis data set would require a lot of work, but could you justify why it is not necessary, and if it is supported by the AWARE observations.**

We have added the following text in new lines 198-204:

ERA-Interim has been superseded by ERA5 (Hersbach et al., 2020) and it is worth considering if differences in the dynamical fields between the two reanalyses potentially give different k-means clustering results. In Figure 3 we compare the 700 hPa geopotential heights between the two for a region over the eastern Ross Sea and RIS. The Pearson correlation between the two is $R = 0.9973$, and there is a slight low bias of order 10 m in ERA-Interim compared with ERA5. This tight consistency between the two reanalysis suggests that the same four clusters would emerge if the work were redone with ERA5.

The new Figure 3 gives this comparison.

*3. Geographical components*

**The paper is missing a map locating the ARM sites and the regions mentioned in the manuscript. It is important for the readers that are not very familiar with those regions, to better understand the links with the meteorological regimes. See for example the figure 1 in Scott et al., 2019, or the figure 1 in Silber et al., 2019.**

A new map with all locations labeled now appears as the new Figure 1.

*Also, the Figure 1 shall contain latitudes and longitudes.*

This figure, now Figure 2, has been improved as requested.

*4. Relevance of the MODIS cloud products for this study*

**In this paper, MODIS is used to assess the cloud phase and cloud top height over McMurdo station during the summer 2015-2016. As mentioned in the manuscript, some improvements have been made in polar regions (Frey, 2008), but inconsistencies still exist for cloud**

*occurrence (e.g., Cossich et al., 2021; Marchant et al., 2016) and cloud properties (e.g., Wilson et al., 2018).*

*The impact of uncertainties in MODIS products on the analysis is an important consideration, especially in the presence of multi-layer clouds or potential omissions of certain clouds. The authors should explicitly address how these uncertainties might affect their findings, particularly in scenarios where the MODIS data might misrepresent the actual cloud conditions.*

We have included the following text in new lines 239-247:

Although MODIS cloud property retrievals have shown improvements over successive data collections, some uncertainties remain over high latitude regions that may never be fully mitigated. Cossich et al. (2021) compared MODIS cloud amount retrieval over the Antarctic Plateau with retrievals from surface-based spectral infrared remote sensing, and found that MODIS consistently underestimated cloud amount by a factor of two or more year-round, although MODIS did reproduce the amplitude of the seasonal cycle in cloud amount. Marchant et al. (2016) note that MODIS cloud top detection becomes less reliable for optically thin clouds at colder temperatures < 240 K, and these associated errors propagate in a cloud phase identification algorithm. Wilson et al. (2018) compared MODIS cloud single-phase microphysical retrievals (optical depth and effective radius) with retrievals from a surface-based shortwave spectroradiometer on the WAIS, and noticed discrepancies likely related to phase detections errors in low clouds close to the snow surface.

*My other comment is why the authors didn't use ground-based observations (radar, lidar, ARM VAP, …) acquired during AWARE to do their analysis or at least assess the cloud properties retrieved by MODIS (see Minor comment below on Figures 2 and 3).*

We have included a new Figure 5 using the ARM THERMOCLDPHASE VAP, along with the following text in new lines 256-270:

For these reasons, we checked the conclusions in Figure 4 against a surface-based ARM Value Added Product (VAP) from AWARE. We use the THERMOCLDPHASE VAP (de Boer et al., 2011) which contains phase identification in up to ten cloud layers from the High Spectral Resolution Lidar (HSRL). Analogous to Figure 4a, Figure 5 shows the occurrence frequency of clear skies and cloud phase classifications for each of the four regimes, for the highest HSRL-detected cloud layer. There is some qualitative consistency with the MODIS results, particularly for Regime 4, which shows the lowest clear sky frequency and highest ice cloud frequency. Also, the warm Regime 1 shows the greatest occurrence of liquid water cloud. The obvious discrepancy is the severe under-detection of liquid water cloud compared with MODIS, and strong preference for ice cloud in all regimes. This inevitably results from the HSRL's upward-looking view. Silber et al. (2021) analyzed AWARE data from multiple sensors and determined that ~75% of supercooled liquid-bearing clouds precipitate ice particles from the cloud base, and this is consistently detected by the upward-looking lidar. Above these same predominantly liquid water clouds, the MODIS cloud phase algorithm would identify most of them as containing mainly liquid water. Despite the discrepancies, both the VAP and MODIS retrievals show a

relative preference for cloud liquid water in the warm regimes and ice water in Regime 4. We now investigate the potential manifestation of these phase contrasts in surface radiometric measurements.

***Minor comments:***

***Page 4, Line 123: In the past, maintaining the measurement quality of some radiometers has been a challenge when used in polar environment, for example when icing appears on the optics (Cox et al., 2021). Could you comment on how feasible it would be to have the suite of instruments you suggest in unattended remote location?***

We have added the following text in new lines 663-669:

There are naturally some limitations if these radiometers are left unattended, in particular, ice accretion and riming on the optics (e.g., Cox et al. 2023). If the radiometers accompany an expedition then presumably a team member can ensure optical cleanliness on a daily basis. Also, if there is sufficient power to ventilate the radiometers with blowers this will prevent most optical fouling except in the harshest conditions. At the same time, unattended shortwave and longwave radiometers have been successfully deployed with Antarctic automatic weather stations and have yielded long time series with useful results when relative values are analyzed or suitable parameterizations are developed to account for periods of icing (van den Broeke et al., 2004).

***Page 8, Table 1: The date January 26, 2016, appears 2 times.***

This has been corrected.

***Pages 8 and 9, Figure 2a and 3: Could you comment on the differences between MODIS clear sky and TSI cloud cover. For example, MODIS identified clear sky during 15%, while TSI observed clear sky for more than 25 %.***

We have added the following text in new lines 280-284:

Comparing Figure 6 with Figure 4a, we notice that MODIS identification of clear skies is 6% larger than TSI in the warmer regimes (1 and 3), and 10-11% smaller than TSI in the colder regimes (2 and 4). The discrepancies for the warmer regimes are quite small considering the widely differing fields of view between the TSI all-sky camera and the MODIS satellite footprint. The discrepancies for the colder regimes may be analogous to those found in Cossich et al. (2021).

***Page 12, Figure 6: The figure may be easier to read using a log-scale.***

A log-scale didn't improve the readability so we left the figure as is.

***Page 13, Line 306: Particle size could be another important parameter influencing the emissivity of optically thin clouds.***

This has been clarified in new line 441.

***Pages 14 and 16: Figures 7 (b, d, f) and 8 (b, d, f): It would be easier to read with the dashed line crossing 0 on both y-axes, and a legend for the red bars and black line.***

These plots (now Figures 10 and 11) have been revised as requested.

---

## Author Response (AR2)

**Manuscript egusphere-2023-1665 – Replies to Reviewer #2 Second Report with Revisions**

**The authors correctly addressed the major comments I had, giving greater depth and significance to their work.**

The reviewer's comments were indeed very helpful toward improving this paper, and we thank the reviewer for the thoughtful review.

I appreciated that they included a discussion of the comparison with the ARM ground-based instruments, showing some disparity with the phase observed by MODIS. However, to my knowledge, the VAP THERMOCLDPHASE products use the HSRL instrument in addition to other observations (KAZR, MWR), which makes it possible to identify the phase of layers beyond the altitude where the lidar signal is extinguished. The authors should, therefore, revise the section starting with "This inevitably results from the HSRL's upward-looking view" (Page 10, Line 254).

This section has been clarified. The point is not that THERMOCLDPHASE would not detect the phase of higher cloud layers beyond the lidar attenuation, but rather that the majority of these clouds precipitate ice below the cloud base, which is entirely missed by the satellite passive imager (MODIS).

**Interestingly, this additional analysis also shows a general underestimation of clear-sky cases with TSI, most marked for Regime 3 (about 10%). Can the authors comment on this difference?**

This section now mentions a known tendency for TSI to sometimes underestimate clear sky fraction due to forward scattering contamination near the position of the direct solar beam.

**Few minor comments and typos:**

**Page 8, Line 197: A space is missing after ERA5.**

This has been corrected.

**Page 9, Line 226: Should be "reanalyses".**

This has been corrected.

**Page 16, Figure 9: The figure may be easier to read using a log scale.**

The figure now uses a log scale.

**Page 20, Line 457: In the PDF version I reviewed, the sentence starting with "considered here, the fact ... " appears two times.**

This has been corrected.